# Extended Cleavage Specificity of the Rat Vascular Chymase, a Potential Blood Pressure Regulating Enzyme Expressed by Rat Vascular Smooth Muscle Cells

**DOI:** 10.3390/ijms21228546

**Published:** 2020-11-12

**Authors:** Petter Berglund, Srinivas Akula, Zhirong Fu, Michael Thorpe, Lars Hellman

**Affiliations:** The Biomedical Center, Department of Cell and Molecular Biology, Uppsala University, Box 596, SE-751 24 Uppsala, Sweden; berglund.o.p@gmail.com (P.B.); srinivas.akula@icm.uu.se (S.A.); fuzhirong.zju@gmail.com (Z.F.); getmeinahalfpipe@gmail.com (M.T.)

**Keywords:** chymase, chymase locus, serine protease, Angiotensin I, Angiotensin II, blood pressure, regulation

## Abstract

Serine proteases constitute the major protein content of the cytoplasmic granules of several hematopoietic cell lineages. These proteases are encoded from four different loci in mammals. One of these loci, the chymase locus, has in rats experienced a massive expansion in the number of functional genes. The human chymase locus encodes 4 proteases, whereas the corresponding locus in rats contains 28 such genes. One of these new genes has changed tissue specificity and has been found to be expressed primarily in vascular smooth muscle cells, and therefore been named rat vascular chymase (RVC). This β-chymase has been claimed to be a potent angiotensin-converting enzyme by cleaving angiotensin (Ang) I into Ang II and thereby having the potential to regulate blood pressure. To further characterize this enzyme, we have used substrate phage display and a panel of recombinant substrates to obtain a detailed quantitative view of its extended cleavage specificity. RVC was found to show a strong preference for Phe and Tyr in the P1 position, but also to accept Leu and Trp in this position. A strong preference for Ser or Arg in the P1’ position, just C-terminally of the cleavage site, and a preference for aliphatic amino acids in most other positions surrounding the cleavage site was also seen. Interesting also was a relatively strict preference for Gly in positions P3’ and P4’. RVC thereby shares similarity in its specificity to the mouse mucosal mast cell chymase mMCP-1, which efficiently converts Ang I to Ang II. This similarity adds support for the role of β-chymases as potent angiotensin converters in rodents, as their α-chymases, which have the capacity to efficiently convert Ang I into Ang II in other mammalian lineages, have become elastases. However, interestingly we found that RVC cleaved both after Arg2 and Phe8 in Ang I. Furthermore this cleavage was more than two hundred times less efficient than the consensus site obtained from the phage display analysis, indicating that RVC has a very low ability to cleave Ang I, raising serious doubts about its role in Ang I conversion.

## 1. Introduction

Serine proteases are expressed by cells of several of the major hematopoietic cell lineages, including mast cells, neutrophils, cytotoxic T cells and natural killer cells (NK-cells) [1]. These proteases, which belong to the large family of trypsin/chymotrypsin-related serine proteases, are stored in their active form within cytoplasmic granules of the cell [1,2,3]. The genes of the hematopoietic chymotrypsin-related serine proteases are in mammals encoded from four different loci, the mast cell chymase locus, the mast cell tryptase locus, the met-ase locus and the T cell tryptase locus [1,4]. The mast cell chymase locus does in primates contain four protease genes, the mast cell α-chymase (HC), cathepsin G (CtsG) and two granzymes (GzmH and GzmB) (Figure 1). Interestingly, this locus has in rodents experienced massive expansions both in size and in the number of active genes. In mice, this locus is approximately 2.5-times larger than the human counterpart whereas the rat locus is 8.6-times larger than the human counterpart and encodes 28 functional genes [4] (Figure 1). Based on phylogenetic analyses of these genes, the chymases can be divided into two groups, α- and β-chymases (Figure 2) [5,6]. The β-chymases are only found in rodents except for one member found in cats (Cat MCP-like) and one in dogs (Figure 1 and Figure 2). Further expansion of the original β-chymase gene is likely to have taken place in rodents as eight β-chymase genes can be found in the rat genome and five in mice (Figure 1 and Figure 2) [1,4]. The fact that β-chymases are found in both rodents and in cats and dogs but not in other mammalian lineages indicates that the β-chymases are relatively old and most likely have been lost in some mammalian lineages [4].

In ruminants such as cattle and sheep this locus has also experienced a relatively large expansion [4]. An additional subfamily has appeared within this locus named duodenases due to their expression pattern (Figure 1). The duodenases are no longer expressed by hematopoietic cells but instead in the small intestine, more specifically in the Brunners glands of the duodenum [7,8]. This new subfamily is most likely the result of gene duplications of cathepsin G or one of the granzymes.

In contrast to the ruminants, the expansion in mice and rats can be attributed to both additional granzymes and chymases [1,4]. Interestingly, in the rat one of these new β-chymases has also changed tissue specificity and is now expressed in vascular smooth muscle cells [9]. This protease has therefore been named rat vascular chymase (RVC). This protease is particularly interesting due to one reason, the potential role of these chymases in blood pressure regulation. The human, the macaque, and the dog chymases and the mouse counterpart, mMCP-4, have all been shown to be potent angiotensin converters by cleavage of a peptide bond between amino acid 8 and 9 in angiotensin I (Ang I) [10,11]. This results in the formation of Ang II, which is a potent blood vessel constrictor, and an increase in blood pressure. Interestingly, rMCP-1, the rat counterpart of the mast cell expressed chymase of mouse mast cells, mMCP-4, may not be a good angiotensin converter. rMCP-1 has namely been shown not only to cleave after Phe8 in Ang I but also after Tyr4, with the latter resulting in inactivation of Ang II [6,11,12,13]. The role of RVC may here be to rescue the loss of function of rMCP-1 as an angiotensin converter.

The renin-angiotensin system is an endocrine system which through the generation of Ang II from Ang I, regulates blood pressure and water and electrolyte balance [14]. The conversion from the decapeptide Ang I to the octapeptide Ang II is done by cleavage of Ang I Phe8-His9 peptide bond [15]. In the circulatory system, the conversion is done by the angiotensin-converting enzyme (ACE), whereas in tissues, the conversion can be done via chymase, without the involvement of ACE [14]. In human vascular tissues, up to 95% of Ang II conversion has been attributed to HC [16]. The α-chymases present in mice and rats (mMCP-5 and rMCP-5) do not convert Ang I to Ang II, due to the fact that they have changed primary specificity from being chymases to now being elastases with a preference for small aliphatic amino acids [17,18]. Possibly to compensate for this change in primary specificity of the α-chymase in mice, the mMCP-4 has been shown to efficiently produce Ang II similar to HC [11]. In rats, the β-chymase rat vascular chymase (RVC) may have rescued this function if the major connective tissue mast cell chymase in rat rMCP-1 also inactivates Ang II [6,11,12,13].

RVC has been shown to be produced primarily in vascular smooth muscle cells and an increased expression has been observed in hypertensive rats [9]. Interestingly also, overexpression of RVC in transgenic mice has been shown to cause hypertension associated with medial thickening of arteries [19]. In addition, silencing of RVC with small interfering RNAs showed reduced Ang II production in vascular smooth muscle cells exposed to high glucose but not in cells with normal glucose conditions [20]. When using two different hypertensive rat models, the chymase inhibitor chymostatin has also been found to reduce Ang II concentrations both in the blood plasma and the kidney, indicating that chymase is involved in Ang II production in both kidney and plasma [21]. The evidence from several independent studies using different models and strategies to study the role of RVC in angiotensin conversion thereby strongly favors a potent role of RVC in angiotensin conversion in rats.

To increase our knowledge of this interesting enzyme we have decided to obtain more detailed information concerning the biochemical characteristics of this enzyme. One such important characteristic is its extended cleavage specificity. Such knowledge could help gain a better understanding of the in vivo function of this new enzyme and its change in tissue expression in order to put the development of chymases as angiotensin converters in an evolutionary context. In this study, the aim was to characterize RVC through determination of its extended cleavage specificity by using phage display technology and to verify and obtain quantitative information concerning the importance of residues at and around the cleavage site by the use of recombinant substrates. We have also used mass spectroscopy (MS) and recombinant substrates to analyze the cleavage of Ang I by RVC and a panel of mammalian mast cell chymases to further study their potential roles in Ang I conversion.

## 2. Results

### 2.1. The Enzymes

Recombinant rat vascular chymase and rMCP-1 were produced in the human embryonic kidney cell line HEK293-EBNA, with an N-terminal His_6_-tag and an EK site (Figure 3A). After purification of the His tagged enzyme on Ni^2+^ chelating IMAC columns from the conditioned cell medium, the enzyme was activated by digestion with enterokinase (EK) (Figure 3B). The gel shows the reduction in molecular weight of the majority of the RVC enzyme after EK digestion, indicating a successful removal of the EK-site and His_6_-tag and the activation of the protease (Figure 3B). The purification and activation of rMCP-1 has been presented in a previous publication [22].

### 2.2. Chromogenic Substrate Assay

To study the primary cleavage specificity of RVC, we used a set of chromogenic substrates. Five substrates were used for an initial analysis of the primary specificity of this enzyme. We used a chymase substrate Suc-AAPF-pNA, a leu-ase substrate Suc-AAPL-pNA, an elastase substrate Suc-AAPV-pNA, a trypsin substrate Suc-VLGR-pNA and asp-ase substrate Ac-VEID-pNA. After addition of RVC, measurements were made at the following time points 0, 20, 40, 60, 120, 180, 240, 300 and 360 min (Figure 3C).

Results show an overall relatively low activity of the enzyme with visible cleavage primarily of the chymase substrate, and a very low activity of the Leu and Arg containing substrates (Figure 3C). The low activity on the Arg substrate is most likely coming from the small amounts of the activating enzyme, the enterokinase. No cleavage of the other substrates can be seen (Figure 3C).

### 2.3. Substrate Phage Display

To determine the extended cleavage specificity of RVC, substrate phage display was performed. The library we used had a complexity of approximately 5 × 10^7^ variants. The T7 phages express one copy each of a modified capsid protein containing a 9 amino acid random region followed by an His_6_-tag to be able to attach the phages to a solid substrate, in this case Ni^2+^ chelating IMAC sepharose beads. After six rounds of selection, the phages selected by RVC had increased by 66 times compared to the PBS control. A total of 120 plaques from this last selection round were picked and an approximately 300 bp region containing the region encoding the random 9 amino acids was amplified by PCR and the PCR products of the 96 samples with most clean PCR fragments were sent for sequencing.

The sequences received were translated and aligned manually based on the primary specificity obtained from the chromogenic substrate assay. Each row in the figure shows the 9 amino acids selected by RVC, with four amino acids upstream and downstream of the predicted cleavage site, the P1 position (Figure 4A). From the phage displayed sequences, the preference for a particular amino acid in each position could be obtained (Figure 4A). A strong preference for Phe and Tyr was observed but also some sequences with a Trp or a Leu in the P1 position was seen. Just C-terminally of the cleavage site, the P1’ position, a strong preference for either Arg or Ser was observed (Figure 4A). In the remaining positions a larger variation of amino acids was observed, however, with a tendency for small aliphatic amino acids such as Val, Leu or Ala. We could also see a preference for Gly in positions P3’and P4’ (Figure 4A). From the phage displayed sequences, the consensus sequence was therefore found to be Val-Leu-Leu-Phe-Ser-Ala-Gly-Gly. An Ice-logo type presentation of this preference of this enzyme is presented in Figure 4B.

The phage display sequences from an earlier study of rMCP-1 were also included in Figure 4 for comparison [22].

### 2.4. Verification of Phage-Displayed Sequences Using the Two-Trx System

To verify the results received from the phage display as well as investigate the effect of single amino acid substitutions on the cleavage efficiency, a two-Trx system was used. In this system, recombinant sequences consisting of 8 amino acids are inserted in the linker region between two *E. coli* Trx molecules with one, the second Trx, having a C-terminal His_6_-tag (Figure 5A). The inserted sequence can be targeted for cleavage by proteases, resulting in a separation of the two Trx molecules (Figure 5B). After adding the protease, aliquots were taken at different time points (0, 15, 45 and 150 min) after which the reaction was stopped by addition of SDS containing sample buffer and 1 uL β-mercaptoethanol. The aliquots were then run on 4–12% SDS-PAGE gels. On the gel, the band at ~28 kDa represents the uncleaved construct and the two bands at ~14 kDa each represent one Trx molecule. A panel of more than 300 such substrates has previously been designed and produced for the verification of the extended cleavage specificity following phage display analysis for a number of previously analyzed mammalian hematopoietic and coagulation serine proteases [10,23,24,25,26,27,28,29,30,31,32].

First, a number of reactions using previously designed substrates for the rat chymase rMCP-2 and HC were done to verify the preferred P1 and P1’ residues (Figure 5C, D) [27]. Here, a preference for Phe and Tyr over Leu and Trp in the P1 position can be seen with Phe being the most preferred P1 residue. In the P1’ position, Ser and Arg are preferred over Leu, with Ser being the most favored amino acid in this position. This confirms the results from the phage display where most of the sequences either have Phe or Tyr in the P1 position and either Ser or Arg in the P1’ position (Figure 4A).

Further analysis of the extended cleavage site was made with the use of the consensus sequence derived from the phage display analysis (VLLFSAGG). In the P2 position, RVC showed lower specificity than what could be seen for the P1 and P1’ positions. However, the cleavage was inhibited by the presence of the negatively charged Glu (Figure 5E).

The amino acids in the P3’ and P4’ positions, C-terminally of the cleavage site, were also found to be of importance. A clear preference for Gly in both of these positions was observed. Substitution of the P3’ Gly with either Val or Arg resulted in a reduction in cleavage. An even stronger inhibition can be seen when both Gly are substituted with larger aliphatic residues, in this case with a P3’ Val and a P4’ Leu (Figure 5F). Only a small change in efficiency in cleavage can also be seen with a Glu or a Gly in the P2’ position, in agreement with the phage display results, which also indicated a low importance of any particular amino acid in the P2’ position (Figure 4A).

### 2.5. Cleavage of Angiotensin I

Although numerous articles highlight the potential role of RVC in blood pressure regulation, no analysis of the cleavage of Ang I by pure recombinant RVC has yet been published. Therefore, to study the activity of RVC on Ang I we have analyzed the cleavage of this enzyme in a two-Trx substrate containing the entire sequence of human Ang I positioned in the inserted linker region. This construct also contains structurally flexible kinker regions (Ser-Gly) surrounding the Ang I sequence to make the sequence more accessible for cleavage (Figure 5A). As a reference substrate we used the consensus sequence obtained by phage display (VLLFSAGG). The amount of enzyme used to visualize cleavage of the Ang I Trx construct was 50 times higher than the one used for the consensus site and still there was quite a lot of uncleaved material in this lane compared to the consensus site (Figure 5H). The results thereby showed that 200–300 times more enzyme was needed (based on having 4–6 times higher amounts of uncleaved substrate) to cleave the Ang I in the 2xTrx setting compared to the consensus substrate, which shows that Ang I is a very poor substrate for this enzyme (Figure 5H).

To determine the exact cleavage sites within Ang I for RVC we sent Ang I peptides (uncleaved, cleaved with HC, cleaved with rMCP-1 and with RVC) for mass spectrometry (MS) analysis (Figure 6). HC cleaved as expected at Phe8, generating Ang II. In contrast, rMCP-1 generated small amounts of Ang II but primarily degradation products by cleavage both at Tyr4 and Phe8, as previously shown by several labs [6,11,12,13]. In addition, to our big surprise RVC cleaved primarily at Arg2 and also to some extent at Phe8, the latter only after using more than 200 times the amount of enzyme compared to the HC and rMCP-1 (Figure 6).

To confirm these results and to study the cleavage of Ang I by a panel of other mast cell chymases we designed a new type of recombinant fusion protein based on the 2xTrx substrate. In this new substrate, an Ang I sequence was inserted after the coding region of the first Trx, followed by a short kinker region with repeating Ser-Gly residues with a six-histidine tag and a single Gly residue just preceding the Ang I sequence (Figure 7A). After the Ang I sequence a stop codon was inserted. The Ang I sequence is thereby sitting at the C-terminus of the fusion protein, which is preceded by a highly flexible kinker region. This fusion protein as well as five different mutants of the human Ang I sequence were analyzed by cleavage, followed by separation on SDS-PAGE gels. As expected, the HC and the dog chymase almost exclusively cleaved after Phe8 in the Ang I sequence but not at Tyr4, exactly as shown by peptide cleavage analyzed by MS (Figure 6B and Figure 7D) [10]. This was confirmed for the HC using two mutants: one where the Phe8 had been mutated to a Ser residue, and a second mutant where the Tyr4 had been mutated to a Ser. The first mutant resulted in no cleavage at all and the second substrate was cleaved equally efficient as the wildtype substrate by HC (Figure 7E). When rMCP-1 was analyzed with these substrates, the substrate was cleaved at Tyr4 (Figure 7C). However, by analyzing the Tyr4/Ser mutant, rMCP-1 also efficiently cleaved at Phe8, showing that cleavage occurred at both sites, similar to what was observed from the MS analysis (Figure 6C and Figure 7F). Analysis of RVC using the wildtype sequence and five mutants of Ang I involving single, double and triple mutants in positions Arg2, Tyr4, Ile5 and Phe8 showed that Arg2 was the primary site for cleavage by RVC (Figure 6D and Figure 7G,H). Minor cleavage was also observed at Phe8 (Figure 6D). This indicated that both Tyr4 and Phe8 were very poor substrates for RVC, probably because of suboptimal surrounding residues. When the aromatic amino acids were surrounding the suboptimal Arg, cleavage occurred after using more than 200 times the amount of enzyme needed in comparison to the optimal consensus site obtained from the phage display analysis (Figure 5H). However, the surroundings of the Arg2 had an effect on the cleavage at this site, as the Tyr4/Ser and the Tyr4/Ser+ Ile5/Ser mutants both showed higher amounts of uncleaved fusion protein compared to the wildtype construct. The Tyr was apparently favored over Ser in the P2’ position when RVC cleaved after Arg (Figure 7G,H). Due to the presence of low amounts of enterokinase in the RVC enzyme preparation as indicated in Figure 3 it is likely that at least part of this Arg specificity originates from the enterokinase. We therefore tested cleavage of the Trx-Ang I substrate with the same amount of enterokinase as was present in the RVC used for the cleavage in Figure 7G and could see a 50% cleavage of the substrate indicating that at least part of the Arg2 cleavage can be attributed to enterokinase.

The cleavage of Ang I in the Trx vector was analyzed by several additional mast cell chymases, from hamster, opossum and platypus, and all of them except the platypus chymase behaved like rMCP-1 by cleaving at both Tyr4 and Phe8 and thereby having strong tendencies to also degrade Ang I (Figure 7I–L). The platypus chymase (Granzyme B) showed a similarity to RVC also a preference for cleavage at Arg2 in Ang I (Figure 7L).

The results from the MS analyses confirmed the findings from the Trx-Ang I fusion protein analyses for the HC, dog chymase, rMCP 1 and RVC, suggesting the latter cleavage reactions were robust and accurate (Figure 7 and [10]).

## 3. Discussion

One of the central questions in the biology of the different hematopoietic serine proteases are their in vivo targets. Several potential targets have been identified for the human chymase and for the corresponding mouse enzyme mMCP-4, including Ang I, various snake and scorpion toxins, fibronectin, fibrinogen, thrombin and several key TH2 cytokines [1,2,3,36,37]. However, the question concerning the role of these mast cell proteases in angiotensin conversion has been questioned by the fact that rMCP-1, the rat counterpart of mouse mMCP-4, has been shown to be a very poor generator of Ang II due to cleavage of Ang I at two sites both after Phe8 and Tyr4. This cleavage at Tyr4, which results in inactivation of Ang II, indicates that angiotensin conversion may not be a major function of the major mast cell chymase. The identification of RVC and its potential potent role in blood pressure regulation was therefore of major interest as this in our mind can be seen as a strong argument for the role of these enzymes in angiotensin conversion and thereby blood pressure regulation. When the major rat mast cell chymotryptic enzyme rMCP-1 may have lost its angiotensin-converting function, another of the rat β-chymases may have taken over this function and interestingly also changed tissue specificity, from mast cells to vascular smooth muscle cells. To our knowledge this is the only example where one of the mast cell β-chymases has changed tissue specificity from primarily being expressed in mast cells to now being expressed in vascular smooth muscle cells. In most other species analyzed, including human, macaque and dog, the mast cell chymases seem to be potent angiotensin converters of the vasculature but then due to the presence of mast cells in the connective tissue surround the vessels or in the vessel wall. In favor of the role of the mast cell chymases in angiotensin conversion is also the finding that up to 95% of the angiotensin-converting activity in human heart and vessels has been attributed to the human mast cell chymase [16].

Upon the analysis of the cleavage of Ang I by rMCP-1 we could verify the previous studies [6,11,12,13]. Both by analysis with LC-MS/MS and by the recombinant Trx fusion protein, rMCP-1 cleaved at both sites generating small amounts of Ang II but primarily degrading Ang I by cleavage at both Tyr4 and Phe8 (Figure 6 and Figure 7). The same pattern was also seen for the hamster and opossum chymases, indicating a common phenomenon among other mast cell chymases. Furthermore, it suggests that the primate and canine mast cell chymases are the ones that differ from the majority of the mammalian chymases. It was interesting to note that RVC was a very poor Ang I converter, as it cleaved Ang I first after adding more than 200 times the enzyme needed for the cleavage of the consensus substrate from the phage display and also that the cleavage primarily occurred at Arg2 and only to a minor extent at Phe8 (Figure 6D and Figure 7G,H). The cleavage at Arg2 resulted in the inactivation of the angiotensin (Figure 6D and Figure 7G,H). Interestingly, a very similar pattern was observed for the platypus enzyme (Figure 7L). It should here be noted that at least part of this Arg2 cleavage originates from the remaining low levels of enterokinase in both the RVC and platypus enzyme preparations. 

Several earlier studies have given strong support for the role of RVC in blood pressure regulation, including the initial cloning of RVC from rat blood vessel smooth muscle cells, the strong overexpression of this enzyme in naturally hypertensive rats and also the studies in mice of transgenic expression of RVC in mouse blood vessel smooth muscle cells [9,19]. These results have recently also been supported by an independent study of the efficiency of rat plasma membrane soluble extract from rat heart in converting Ang I and Ang 1–12 into Ang II [38]. This effect was almost completely inhibited by chymostatin, indicating a classical chymase [38]. Therefore, several independent groups have found strong indications for the role of RVC in angiotensin conversion in blood vessels and the heart.

As one step in the analysis of the role of this enzyme in angiotensin conversion we here contribute by delivering a more detailed biochemical characterization of this enzyme. The question we have asked is primarily how this enzyme is related in its extended cleavage specificity to the other rodent β-chymases. In this study, RVC was found to show a strong preference for Phe, with a slightly lower preference for Tyr, and even lower for Leu or Trp in the P1 position. Recombinant substrates were then used to obtain quantitative information concerning amino acid preference for several of the residues surrounding the active site. We could here verify the phage display in that Phe is the preferred P1 residue, followed by Tyr and then Leu and Trp. RVC also has a strict preference for Ser and Arg, in the P1’ position. The activity was greatly reduced by introducing a Leu in this position. The P1 and P1’ positions thereby share a very similar specificity to the mouse mucosal mast cell enzyme, the β-chymase mMCP-1 [39]. Both enzymes prefer Phe over Tyr in the P1 position. The same is true with Ser or Arg in the P1’ position. It also shares similar specificity with its closest rat relative rMCP-2, to which it shares an 89% amino acid identity [40], with the primary exception that RVC has a more strict preference for amino acids in the P1’ position [27]. Other positions were not as specific, but aliphatic amino acids were frequently found in all positions except P1 and P1’. In the P2´position RVC showed similar activity for Leu, Arg and Ser, and slightly lower activity for the negatively charged Glu. The similar activity with Leu, Arg and Ser, as well as the activity with Glu still being relatively high, indicates that RVC seems to accept a variety of amino acids in this position. The amino acids in the P2’ position are important for several other mammalian chymases including HC, the mouse β-chymase mMCP-4 and the hamster chymase HAM1, which all prefer negatively charged amino acids in this position [22,31,41]. This was obviously not the case with RVC. For positions P3’and P4’ a preference for small non-charged amino acids was observed.

RVC thereby shows major similarities in extended cleavage specificity to rMCP-2 and mMCP-1 and less so to rMCP-1 [27,31,39]. Two notable differences between RVC and mMCP-1 when compared to rMCP-1 and rMCP-2 are different preferences for residues in positions P1’, P3’ and P4’. In the P1’ position, RVC and mMCP-1 is more specific than rMCP-1 and rMCP-2, with less occurrence of aliphatic amino acids and hydrophobic amino acids other than Ser as observed from phage display data. A preference for smaller amino acids in positions P3’ and P4’ was also observed, where Gly is preferred by RVC over Val and Leu, and this is also the case with mMCP-1 [27,31,39]. By the analyses of two two-Trx substrates used for the analysis of the specificity of RVC we could verify one major difference observed from the phage display analysis of RVC and rMCP-1. RVC does not like Trp in the P1 position, whereas rMCP-1 readily accepts this amino acid in this position (data not shown).

Rodents have two different mast cell populations, connective tissue mast cells and mucosal mast cells, which both express chymases. mMCP-1 is the only active chymase expressed in mucosal mast cells of mice [37]. In contrast, rat mucosal mast cells express several chymases, including rMCP-2, rMCP-3 and rMCP-4 [1,40,42]. The results from this study raise the question of whether the differences in specificity between mMCP-1 and rMCP-2 as well as differences in tissue location of mMCP-1 and RVC have any impact on their role in angiotensin conversion between mouse and rat. Interesting also is the fact that angiotensin conversion in rodents seems to be performed by both mucosal and connective tissue mast cell chymases. This in marked contrast to primates where their mucosal mast cells lack any chymotryptic activity. Only human connective tissue mast cells express the HC, a situation that needs to be taken into consideration when comparing the role of these chymases in angiotensin conversion when using rodents as animal models.

Interestingly, rMCP-1 most likely plays only a minor role in Ang I conversion due to its potent cleavage at both Tyr4 and Phe8 as shown here and in several previous studies (Figure 6 and Figure 7) [6,11,12,13]. An observation should also be added about the apparent very minor role of RVC in Ang I conversion. The total absence of Ang II from the Ang I sample cleaved with RVC upon MS analysis shows that RVC seems a highly unlikely candidate for the generation of Ang II in rats (Figure 6D). This result was also later confirmed by a panel of Ang I mutants in the Trx system (Figure 7). The large discrepancy between these in vitro studies on Ang II generation by these enzymes and the in vivo data from cloning, transgenics and inhibitor studies shows that much remains to fully understand the role of these two (and potentially other) enzymes in blood pressure regulation in the rat. Two previous studies are in line with our finding that RVC is a very poor angiotensin converter: A study by Kirimura et al. as early as 2005 shows that a chymase inhibitor has no effect on blood pressure in hypertensive SHR rats, whereas an ACE inhibitor completely suppressed Ang II formation [43]. In this study they also show that RVC is not detected in the aorta of the hypertensive rats and not in WKY rats, rather only in the lungs of monocrotaline-induced pulmonary hypertensive rats [43]. A second study by Takai et al. shows that chymase inhibitors do not affect Ang II levels but improves vascular dysfunction and survival in stroke-prone spontaneously hypertensive rats [44].

The broader analysis of Ang I conversion by mast cell chymases from a diverse set of different mammals also indicated that Ang I conversion by mast cell chymases is primarily seen in primates and dogs, and possibly other related mammals not yet studied, but may only play a minor role in rats, hamsters, opossums and the platypus (Figure 6 and Figure 7). A potential difference in the timing of the cleavage at Phe8 and Tyr4 may result in a generation of Ang II that is only there for a short time until cleavage occurs at Tyr4, which can partly be seen in the LC-MS/MS analysis of Ang I cleavage by rMCP-1 (Figure 6C). One possible scenario is that the Ang II that may have been produced can bind to a receptor and thereby escape degradation. This timing is very difficult to study in vitro but may have a role in vivo. Before ruling out the role of these mast cell chymases in Ang II generation, this possibility should be kept in mind. In addition, the lack of true mast cell chymase in both rabbits and guinea pigs also raises doubts about the more general role of mast cell enzymes in Ang II generation in vertebrates [45]. An interesting question is therefore if rabbits, and possibly also guinea pigs, use other enzymes from other loci to generate Ang II or if ACE has a more active role in Ang I conversion in this species.

In summary, we can conclude that in spite of the numerous studies and numerous species analyzed regarding the role of mast cell chymases in Ang II generation and their role in blood pressure regulation, this is still a relatively open question. The role of the mast cell chymases in primates in Ang II generation seems to be relatively well established and possibly also in dogs. However, in other mammals, including rats, hamsters, opossums and platypus, as well as rabbits and guinea pigs, the evidence is still relatively weak, indicating that there are major differences among different mammalian species in the role of mast cell enzymes in blood pressure regulation.

## 4. Materials and Methods

### 4.1. Enzyme

Inactive recombinant RVC containing an N-terminal His_6_-tag and an enterokinase (EK) site was produced in a mammalian expression system, using the episomal vector pCEP-Pu2 and the cell line HEK293-EBNA. The enzyme was activated by digestion with EK at 37 °C for 5 h (Roche enterokinase from Sigma-Aldrich St. Louis, MI, USA). The digestion was verified on a 4–12% SDS-PAGE gel (Invitrogen, Carlsbad, CA, USA) after which the activated enzymes were stored at 4 °C until use.

### 4.2. Chromogenic Substrate Cleavage

To determine the primary specificity of RVC, five chromogenic substrates were tested for their sensitivity to cleavage by RVC. We used the following five substrates for this analysis: Suc-AAPF-pNA, Suc-AAPL-pNA, Suc-AAPV-pNA, Suc-VLGR-pNA and Suc-VEID-pNA from Bachem (Bubendorf, Switzerland) and Chromogenix (Mölndal, Sweden). Reactions were prepared in 96 well microtiter plates, to which 5 µL of substrate (0.2 mM final concentration), 10 µL activated enzyme and PBS were added to a final volume of 200 µL. The reaction was done at 20 °C with measurements taken spectrophotometrically with a Versa-max microplate reader (Molecular Devices, Sunnyvale, CA, USA) at 405 nm at 0, 20, 40, 60, 120, 180, 240, 300 and 360 min. Reactions were done in triplets together with a blank to which no enzyme was added. Results were then graphed by subtracting the blank measurement at each time point and using the mean of the three reactions.

### 4.3. Substrate Phage Display

To determine the extended cleavage specificity of RVC, a library of T7 phages expressing one copy per phage of a 9 amino acid long random sequence followed by a His_6_-tag on the capsid protein 10 was used. The library of 10^9^ plaque forming units (pfu) containing approximately 5 × 10^7^ variants of the 9 random amino acids were added to 125 µL Ni-NTA agarose beads (Qiagen, Hilden, Germany) and bound to the beads by rotating the tubes at 4 °C for 1 h. Unbound phages were removed by washing the beads with 1 mL PBS Tween 0.05% + 1 M NaCl 10 times followed by washing twice with 1 mL PBS. The cleavage was made by resuspending the beads in 375 µL PBS and adding 4 µL activated RVC and incubating the mixture rotating at 37 °C for 2 h. A control reaction was prepared with adding PBS instead of the protease. During this incubation, phages expressing sequences preferred by RVC are released from the Ni-NTA beads. After cleavage, the mixture was centrifuged and the supernatant collected, from which 30 µL was collected and diluted in LB+Amp for a ten-fold dilution series for visualization and counting of plaques. From the dilution series, 100 µL of diluted phages was added to 100 µL 0.1 M IPTG and 300 µL BLT5615 *E. coli* (OD_600_ 0.5). To plate the phages, 2.5 mL 0.6% top agar (55 °C) was added and the mixture was poured onto LA-Amp plates (50 µg/mL). Incubation at 37 °C for 2 h was done until plaques could be counted.

To prepare phages for further selection rounds, 100 µL 0.1 M IPTG was added to a culture of 10 mL BLT5615 (OD_600_ 0.5) and incubated for 30 min at 37 °C, to which the remaining supernatant containing released phages was added. This was incubated for 75 min at 37 °C to allow lysis of the bacteria. Bacterial debris was removed by centrifuging 1.5 mL of the solution at 10,000 rpm for 3 min. A tube was prepared with 100 µL PBS and 100 µL 5 M NaCl, to which 800 µL of supernatant was added. This was used as a library for the following day. After each selection round, plaque numbers were compared with the PBS control to monitor the progress. Five more selection rounds were performed with 15 instead of 10 washes of 1 mL PBS Tween 0.05% + 1 M NaCl. After the final selection round, 120 plaques were picked and placed in a phage lysis buffer (20 mM Tris, 100 mM NaCl, 6 mM MgSO_4_ at pH 8.0), then were shaken for 30 min at 4 °C and after which they were stored at 4 °C until use.

### 4.4. Sequencing and Alignment

To acquire the amino acid sequence preferred by RVC, amplification by PCR of the region containing the 9 random amino acids was made using lysed phages from the 6th selection round as a template. The quality of PCR product was analyzed on a 1.2 % agarose gel. Samples where the PCR product contained a clearly visible PCR fragment were loaded onto a 96 well plate and sent for purification and sequencing to GATC Biotech (Sequencing Centre, Cologne, Germany). The sequences received were translated using CLC viewer and aligned manually using Adobe Illustrator.

### 4.5. Verification of Phage Displayed Sequences

To verify sequences received from the phage display data and to investigate how single amino acid substitutions affect the efficiency of cleavage, a system developed in the laboratory was used. In this system, an amino acid sequence determined from the phage display is introduced in the linker region between two thioredoxin (Trx) proteins, one of which has a C-terminal His_6_-tag for purification purposes.

A pET-21 vector containing the two Trx protein sequences with a long insert between them was used as starting vector. Isolation of the long insert plasmid was done using the E.Z.N.A plasmid Miniprep Kit I (Omega Bio-tek, Inc., Norcross, GA, USA). The long insert was removed using SalI and BamHI restriction enzymes. After this, the reaction was run on 0.7% agarose gel and empty plasmids were extracted from the gel using the E.Z.N.A Gel Extraction Kit (Omega Bio-tek, Inc., Norcross, GA, USA). Both 5’ and 3’ oligonucleotide-encoding sequences determined from the phage display were synthesized (Sigma-Aldrich, St. Louis, MI, USA) and the double stranded oligonucleotides were ligated in the linker region between the two-Trx sequences in the empty plasmids. Transformation of competent Top 10 *E. coli* cells was then done using the ligated plasmids plated onto LA-Amp plates (50 µg/mL of Amp) and left overnight at 37 °C. Colonies were then picked and plasmids were isolated using the E.Z.N.A plasmid Miniprep Kit I (Omega Bio-tek, Inc., Norcross, GA, USA) and the ligation was verified by running the plasmids on a 1.2% agarose gel.

After sequencing of the linker region by GATC Biotech (Sequencing Centre, Cologne, Germany) and confirmation that plasmids from the Top 10 bacteria contain the desired sequences, the samples were transformed into competent Rosetta-gami *E. coli*, plated onto LA-Amp plates (50 µg/mL) and incubated overnight at 37 °C. After picking colonies, protein expression could be done. A 10 mL overnight culture of Rosetta-gami *E. coli* containing the desired plasmids was prepared. The O/N culture was added to 90 mL LB+Amp, to which 0.5 mL 20% glucose had been added. This culture was then incubated at 37 °C under vigorous shaking. After 1 h, when the OD_600_ had reached 0.5, 1 mL 100 mM IPTG was added to the culture to induce protein expression from the plasmid. The culture was then incubated at 37 °C, under vigorous shaking for another 3 h. Then, 50 ml of the culture was transferred to a 50 mL falcon tube and pelleted by centrifugation for 12 min at 3000 rpm at 4 °C. The supernatant was discarded, and the rest of the induced culture was added to the same tube and centrifuged again for 12 min at 3000 rpm, at 4 °C followed by discarding the supernatant. The pellet was resuspended in 2 mL PBS and divided into three microcentrifuge tubes. To lyse the resuspended bacteria, sonication was done five times for 30 s with resting 30 s on ice in between to not overheat the sample. This was followed by centrifugation for 3 min at 13,000 rpm at 4 °C after which the supernatant was transferred to a new tube. To purify the proteins, 250 µL Ni-NTA slurry (50% slurry concentration) (Qiagen, Hilden, Germany) was added to each tube and rotated slowly at 4 °C for 45 min. The solution was then transferred to a 2 mL column with a glass filter. The column was then washed once with 1 mL PBS + 0.05% Tween + 20 mM imidazole and then twice with 2 mL of the same solution. Elution of the proteins was done in 6 fractions with PBS + 0.05% Tween + 100 mM imidazole. The first fraction was 100 µL followed by five (fractions 2–6) fractions of 200 µL each. Fractions were run on a 4–12% SDS-PAGE gel (Invitrogen, Carlsbad, CA, USA), after which fractions containing high concentrations of protein were pooled.

To determine the cleavage efficiency of RVC to the phage displayed sequences present in the linker region of the two-Trx system, 25 µg of two-Trx proteins was added to microcentrifuge tubes with PBS to a total volume of 50 µL. After the addition of 2 µL activated RVC, the reaction was incubated at room temperature (20 °C). Aliquots of 10 µL were taken and the reaction was stopped at four different time points by addition of 2.5 µL 4× LDS sample buffer. The different time points used were 0, 15, 45 and 150 min respectively. After the last time point, the samples were run on a 4–12% SDS-PAGE gel (Invitrogen, Carlsbad, CA, USA).

### 4.6. Angiotensin Cleavage

To determine the cleavage products after cleavage of Ang I peptide by three enzymes (HC, rMCP-1 and RVC), 10 ug of a synthetic Ang I peptide (GeneCust, Laboratorie de Biotechnologie du Luxembourg S.A, Dudelange, Luxenbourg) was cleaved with the respective enzymes for 2.5 h at 37 °C. The samples were then sent to Alphalyse A/S in Denmark (Alphalyse A/S, Odense, Denmark) for analysis by LC-MS and LC-MS/MS. Four samples were analyzed: one uncleaved as a control and three cleaved with the enzymes (HC, rMCP-1 and RVC). Each of the generated peptides were then analyzed for their exact molecular weight by LC-MS.

As a second part of this analysis, the Ang I sequences and several different mutants were inserted in the Trx system. This was the same Trx system used for the verification of the consensus cleavage site obtained for the phage display. However, instead of using two Trx molecules, the Ang I cleavage was analyzed by having the Ang I sequence positioned C-terminal of the first Trx molecule so that it was exposed with its C-terminal end free (Figure 7). The size difference between cleaved and uncleaved is thereby relatively small, only a few amino acids. To confirm the exact cleavage site, we therefore used a panel of mutants where the Phe8, the Tyr4, the Arg2 and the Ile5 were mutated into Ser residues. The samples were run on a 4–12% Nu-PAGE SDS-PAGE gels (Invitrogen, Carlsbad, CA, USA) and stained overnight in colloidal Coomassie staining solution and de-stained for several hours according to previously described procedures [46].

## Figures and Tables

**Figure 1 ijms-21-08546-f001:**
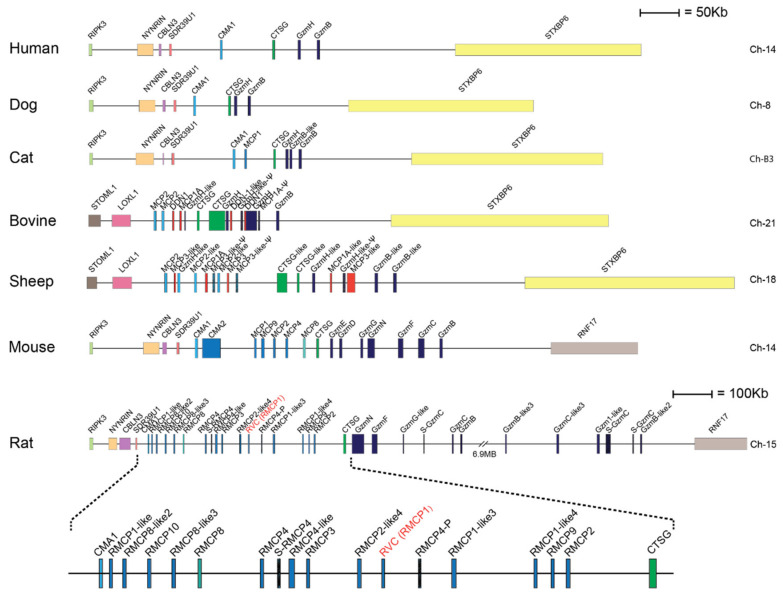
The mast cell chymase locus with their bordering genes of seven different mammals. The species is given to the left side of the figure and the chromosome on which the chymase locus is located to the right. The genes for the serine proteases are shown at double height for more easy identification with their names as found in the database. Granzymes are color coded in dark blue, α-chymases in light blue and β-chymases as blue with a darker tint. MCP-8-related genes are in cyan, duodenases in red and Cathepsin G in green. The location of rat vascular chymase is indicated with red text.

**Figure 2 ijms-21-08546-f002:**
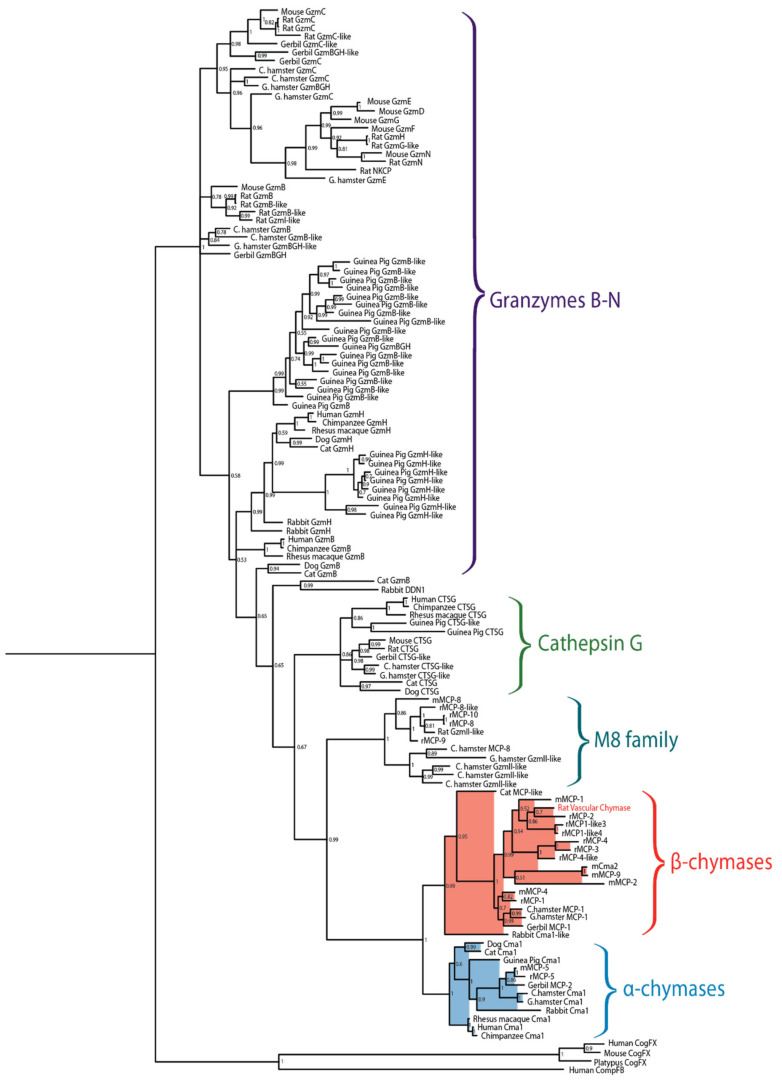
Phylogenetic tree of a panel of mammalian chymases. Sequences originate from human, chimpanzee, rhesus macaque, dog, cat, mouse, rat, gerbil, rabbit, guinea pig, Chinese hamster and golden hamster. The tree was made using the MrBayes algorithm and bootstrap values are displayed at each node. Branches of the mast cell chymases are shaded, with α-chymases in blue and β-chymases in red.

**Figure 3 ijms-21-08546-f003:**
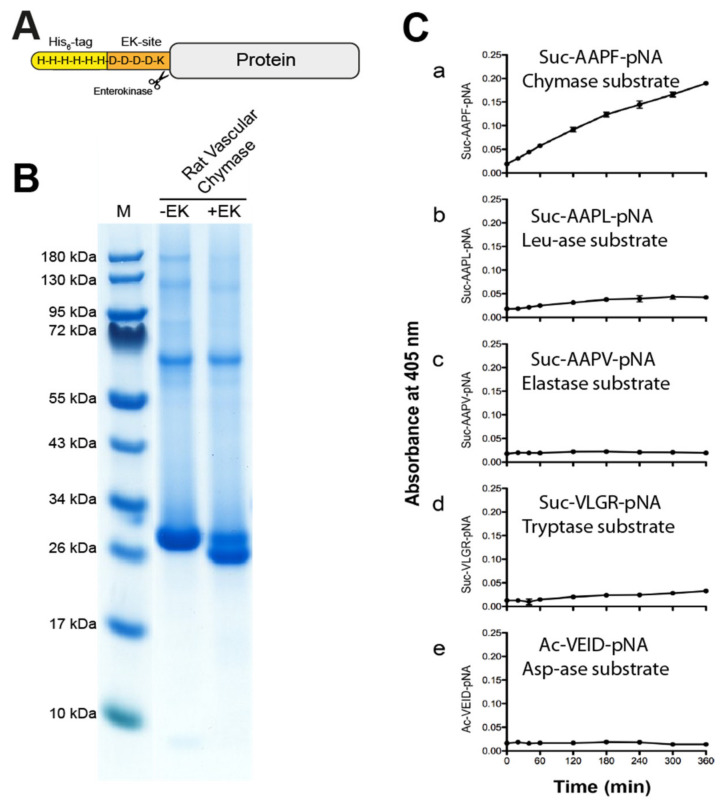
The recombinant rat vascular chymase. (**A**) A schematic drawing of the protein construct. The enzyme was produced inactive with an N-terminal His_6_-tag, and activation was done by digestion with enterokinase (EK). (**B**) SDS-PAGE gel of the protein before and after EK activation. Successful activation results in a drop in molecular weight (right lane). Inactive (−EK) and activated (+EK) protein was run on a 4–12% SDS-PAGE gel. (**C**) Chromogenic substrate assay using RVC (rat vascular chymase) with five different substrates. Reactions were done in triplicate and absorbance measurements at 405 nm were taken at different time points shown on the X-axis. Data points represent mean absorbance of these triplicates with standard deviation. Amino acid sequences of each substrate are shown in each panel.

**Figure 4 ijms-21-08546-f004:**
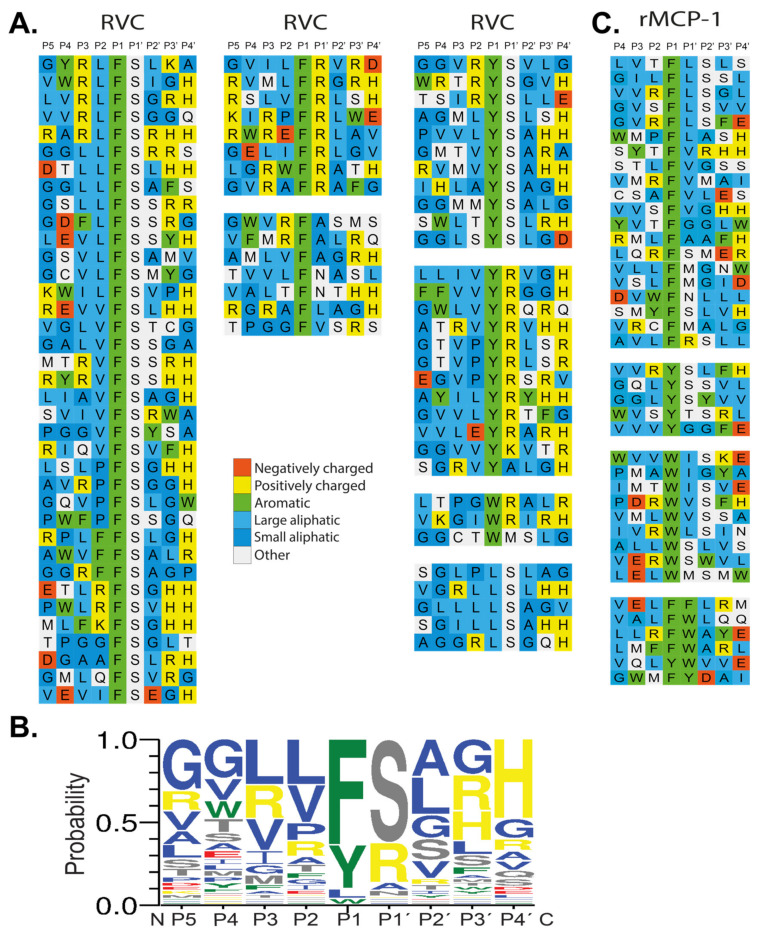
(**A**) Phage display analysis of RVC and rMCP-1. After six selection rounds, plaques were picked and the random nonamer region (PGG(X)_9_HHHHHH, with X indicating randomized amino acids was amplified using PCR. The resulting sequences were aligned P5-P4’ with cleavage occurring between the P1 and P1’ positions. Amino acids are color coded according to their properties as indicated in the figure. (**B**) At the bottom of the figure, we show an Ice-Logo type presentation using the Web-logo program of the distribution of amino acids in positions P5 to P4’ based on the phage display alignment by RVC after six rounds of selection [33]. (**C**) Phage display sequences for rMCP-1 are shown for direct comparison with the sequences obtained for RVC. The phage display sequences for rMCP-1 originate from an earlier publication [22].

**Figure 5 ijms-21-08546-f005:**
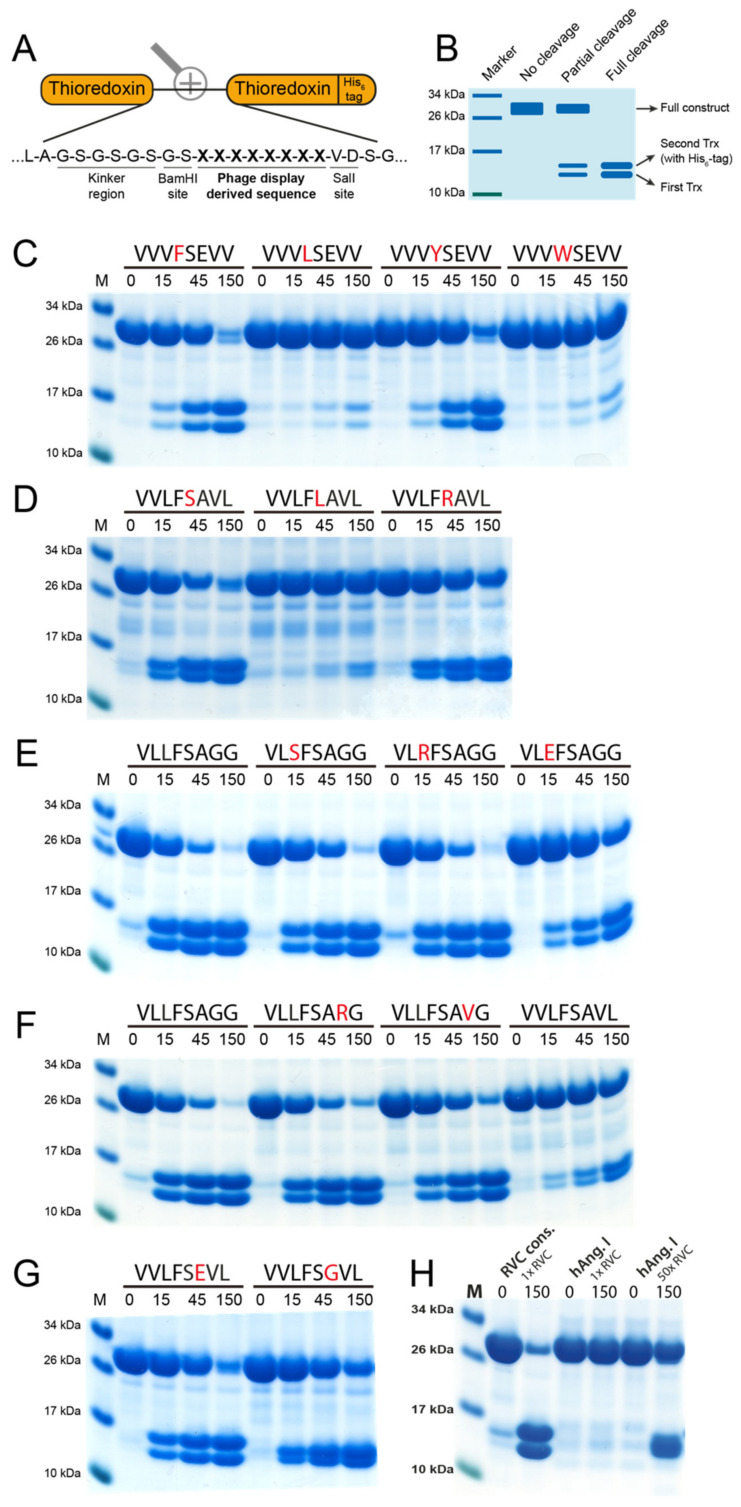
Analysis of the consensus cleavage site of RVC using 2xTrx recombinant substrates. (**A**) Sequences derived from the phage display were inserted in the linking region between two *E. coli* thioredoxin (Trx) proteins, one of which has a C-terminal His_6_-tag. (**B**) Bands at ~28 kDa represent uncleaved two-Trx protein and the two bands present at ~14 kDa represent cleaved protein, with the upper of the two bands representing the Trx containing the His_6_-tag. Sequences above the panels represent the inserted octamers and numbers represent the reaction time in minutes. (**C**,**D**) display cleavage of substrates containing substitutions in positions P1 and P1’. Panels (**E**,**F**) show cleavage of the derived consensus sequence with substitutions in positions P2 and P3’, respectively. (**G**) Position P2’ was investigated using two different substrates. (**H**) The difference in amount of enzyme (RVC) needed to cleave Ang I in the 2xTrx context compared to the consensus site originating from the phage display (VLLFSAGG).

**Figure 6 ijms-21-08546-f006:**
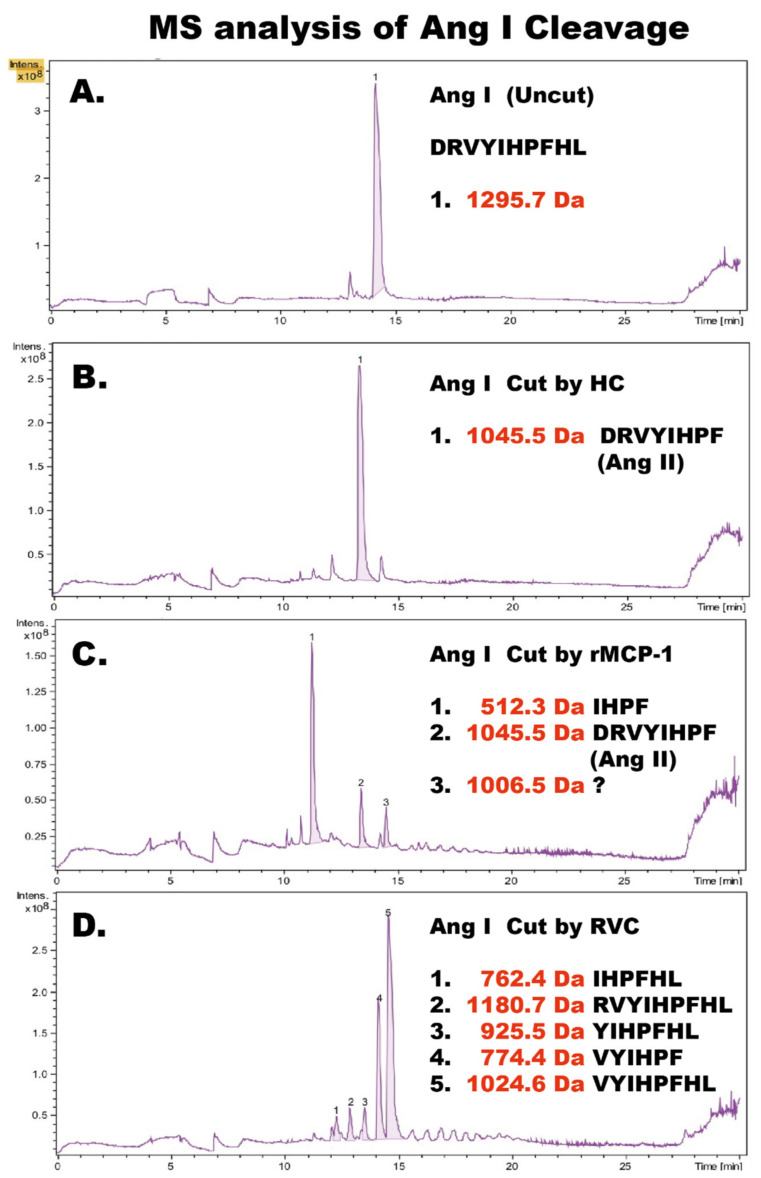
Angiotensin I cleavage. Human angiotensin I was analyzed by LC-MS/MS before cleavage as a control and after cleavage by the human chymase (HC), rMCP-1 and RVC. The cleavage products were then further analyzed for their exact size by LC-MS. The different peaks for each analysis are numbered from 1 to 5 and the exact sequence and molecular weights are listed to the right side within each panel. (**A**) uncleaved Ang I peptide. (**B**) Ang I peptide cleaved with HC. (**C**) Ang I peptide cleaved with rMCP-1. (**D)** Ang I peptide cleaved with RVC. The recombinant enzymes rMCP-1 and the human chymase used in this analysis have been described previously [10,22].

**Figure 7 ijms-21-08546-f007:**
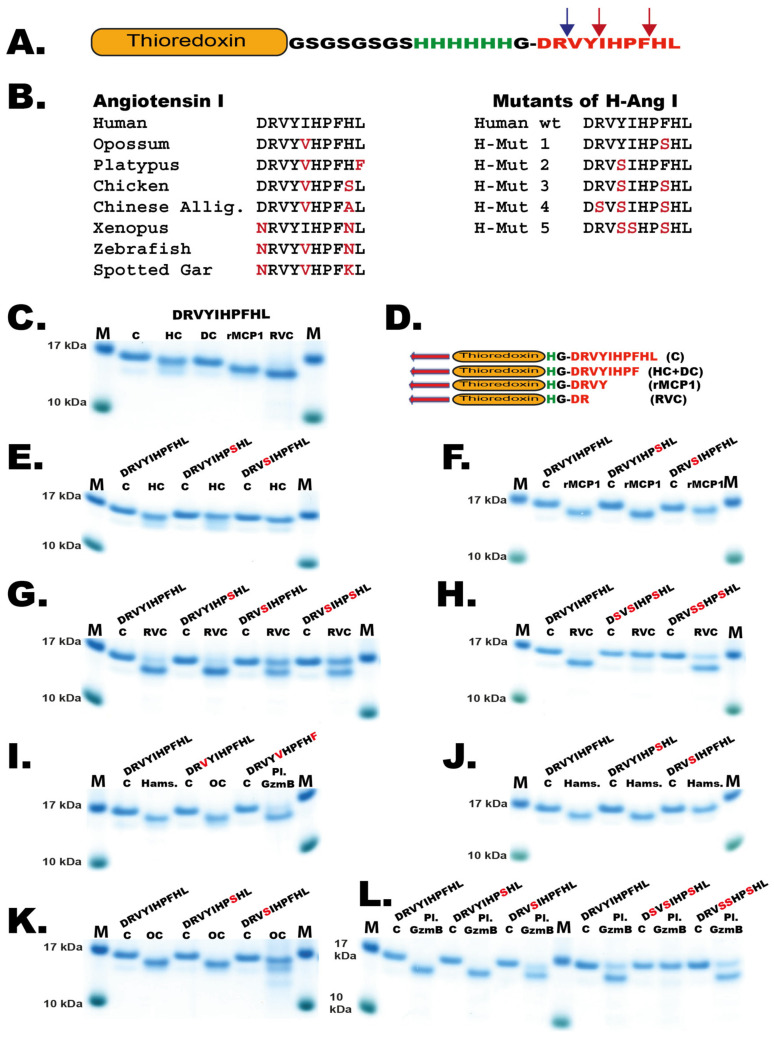
Angiotensin I cleavage. (**A**) The different angiotensin sequences were inserted after a Trx coding region to increase the size in order to be able to see the protein on SDS-PAGE gels. The angiotensin sequence is preceded by short flexible kinker regions consisting of Ser-Gly residues, a six residue His tag and a single Gly for flexibility. The positions of Tyr4 and Phe8 are indicated by red arrows and the Arg2 by a blue arrow. (**B**) Angiotensin I sequences from a panel of mammals, reptiles, amphibians and fish. To the right of these sequences the sequences of five mutants of the human Ang I sequence used to define the exact cleavage sites within the Ang I sequence for the different enzymes are depicted. (**C**) Cleavage of Ang I by the human mast cell chymase (HC), dog chymase, rMCP-1 and RVC. (**D**) The sequences of the different cleavage products generated by these enzymes. (**E**) Analysis of the cleavage of Ang I wt and two mutants Phe8/Ser and Tyr4/Ser by the human chymase. (**F**) Analysis of the cleavage of Ang I wt and two mutants Phe8/Ser and Tyr4/Ser by the rMCP-1. (**G**) Analysis of the cleavage of Ang I wt and two mutants Phe8/Ser, Tyr4/Ser and the double mutant by RVC. (**G**) Analysis of the cleavage of Ang I wt and two mutants Phe8/Ser, Tyr4/Ser and the double mutant by RVC. (**H**) Analysis of the cleavage of Ang I wt and two triple mutants Arg2/Ser + Tyr4/Ser + Phe8/Ser and Tyr4/Ser + Ile5/Ser + Phe8/Ser by RVC. (**I**) Analysis of the cleavage of Ang I wt by hamster chymase, opossum chymase and platypus granzyme B. Platypus granzyme B is the chymase equivalent. The Ang I sequences for opossum and platypus were species-specific sequences. (**J**) Analysis of the cleavage of Ang I wt and two mutants Phe8/Ser and Tyr4/Ser by the hamster chymase. (**K**) Analysis of the cleavage of human Ang I wt and two mutants Phe8/Ser and Tyr4/Ser by the opossum chymase. (**L**) Analysis of the cleavage of human Ang I wt and two single mutants Phe8/Ser and Tyr4/Ser and two triple mutants Arg2/Ser + Tyr4/Ser + Phe8/Ser and Tyr4/Ser + Ile5/Ser + Phe8/Ser by the platypus granzyme B (the chymase). The cleavage of respective Ang I sequence was performed with the recombinant enzyme for 150 min. The proteins before and after cleavage were separated on a 4–12% SDS-PAGE gel. The recombinant enzymes the human chymase, the dog chymase, rMCP-1, the opossum chymase and platypus granzyme B used in this analysis have been described previously [10,22,23,34,35].

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
