# Peer review of "Extended Cleavage Specificity of the Rat Vascular Chymase, a Potential Blood Pressure Regulating Enzyme Expressed by Rat Vascular Smooth Muscle Cells"

_ijms, 2020, doi:10.3390/ijms21228546_

Round 1

Reviewer 1 Report

In this article authors investigate the cleavage specificity of rat vascular kinase that is, in contrast to other chymases, not expressed in hematopoietic cells but is expressed exclusively by rat vascular smooth muscle cells. The role of this kinase is to cleave angiotensin I active angiotensin II which is capable to regulate blood pressure.

From the presented work it is clear that the authors are an experienced team in the field of protease research and the work is done carefully and responsibly. The authors also clearly present the reasons why this chymase is the subject of their interest and tries to clarify its specific position among other serine proteases and at the same time compare its specificity with similar enzymes in other organisms (mice an humans), which are, however, expressed by hematopoietic cells.  

I have only minor comment to the presented work. Figure 3 is a bit confusing to me. The presentation of SDS-page gel occupies almost all the entire image, and the presentation of the chromogenic substrate assay is a bit confusing. It might help if the graphs were larger, or  all substrates could be presented in color in one larger graph. Also, the names of the substrates are not easy to read. Alternatively, for individual graphs, state not only the names of the substrate, but also the name of the enzyme that cleaves the respective substrate (chymase substrate, elastase substrate, ...), so that the image becomes meaningful in itself.

Overall I find the presented article to be well-edited and straightforward.

Author Response

Reviewer 1. The sequence and specificity of the different chromogenic substrates have been added to each panel of the figure of the chromogenic substrate assay.

Reviewer 2 Report

The experiments examining subsite amino acid preferences are fine but very limited in relevance. I had a big problem with the study in that the authors made no attempt to show that their recombinant enzyme could actually cleave angiotensin 1. There in no direct historical evidence for this RVC activity, despite a very misleading statement in the abstract. The introduction only details circumstantial indirect evidence (tissue distribution, overexpreesion in mice etc) that the enzyme might have this function. 

Strangely they get a preferred consensus sequence, but then make no comparisons with the supposed physiological substrates Ang 1 or Ang 1-12. (Ang 1 would not even have a P3’ or P4’ !!).  

So if considering accepting, the authors would need to show the angiotensin cleavage activity and provide kinetic data, as was done in reference no. 11 for dog, human and mouse chymases.  

Quite a few other issues e.g. no source of clone given.

Author Response

Reviewer 2. We agree that a cleavage analysis of Ang I by RVC was needed as this has not been performed by a purified enzyme but only with tissue extracts. We then also included several additional mammalian mast cell enzymes encoded from the chymase locus as reference. To our big surprise we could not verify the results from several previous studies of the cleavage activity by the major rat connective tissue mast cell chymase, rMCP-1. It has previously been shown to both activate and inactivate Ang I by cleavage at both Tyr4 and Phe8. However, we found that pure recombinant rMCP-1 did only cleave at Phe8 as the absolute majority of other mammalian mast cell chymases. We performed a detailed analysis of rMCP-1 and RVC and compared it with a panel of other mammalian mast cell chymases resulting in two new figures 6 and 7. RVC was in contrast to rMCP-1 shown to cleave at both sites, both Tyr4 and Phe8 indicating that it may be less potent as Ang I converting enzyme. We are thankful for the reviewer to give the suggestion to look deeper into the Ang I cleaving activity of RVC as it has resulted in a number of new unexpected findings including the Ang I cleaving activity of rMCP-1, RVC, the Guinea pig Leu-ase and also opossum chymase.

To better adhere to the new data we have changed the title of the manuscript, added an entire section to the results and several sections to the discussion as well as two new figure with figure legends.

Reviewer 3 Report

In this manuscript, Berglund and colleagues investigated the specificity and the cleavage activity of the rat vascular chymase through different approaches. The authors compared the activity of the rat vascular chymase with the activity of other rat and mouse chymases particularly in the regard of the capacity to convert angiotensin I into angiotensin II.

The manuscript is well written and the experiments were correctly scheduled and conducted.

I would just to suggest the authors to add a couple of sentences to discuss more in detail how their data could be translated into human settings, and in the meanwhile to short the introduction and discussion that appear in parts redundant. This could help in better understanding the relevance of the work.

Author Response

Reviewer 3. We have added a new section in the end of the discussion to address the question of the conservation of Ang I cleaving activity in mammals, which gives additional support for the role of Ang I cleavage by the human chymase and thereby of mast cell chymases in blood pressure regulation.

Round 2

Reviewer 2 Report

The major issue this reviewer had with the initial study, was that the authors failed to show angiotensin converting activity with their recombinant protein, and the statement that this was prior knowledge was misleading.

In the revised manuscript the authors have included a section (2.5) Cleavage of Angiotensin 1, seeking to address this. However, they have still failed to examine cleavage of Angiotensin 1. Instead they have inserted the Ang 1 peptide sequence into their two-Trx system, whereby the peptide is part of a large polypeptide chain with flanking Trx sequence. This is not the same as a free peptide substrate – there may be significant differences in conformation and peptide bond accessibility. The “so called” flexible Ser-Gly linker appears to be at one end only, and regardless, this still does not resemble the free N-terminal of the Ang 1 peptide.

Why not assay for cleavage of the actual Angiotensin 1, a readily available substrate?

Given that these new results obtained for RVC and rMCP-1 were unexpected and out of line with 4 previous studies, then this reviewer has to insist that a real and relevant Ang 1 to Ang II assay is conducted and cleavage sites determined. Methods are described in numerous publications (e.g. your ref 11, Caughey et al). Kinetic data should be obtained and compared with previous literature.

A poorly explained part of the study is the original design of the Trx-system using a consensus sequence from the phage display results. This 8aa sequence with a centered cleavage site (P4 – P4’) bears no resemblance to Angiotensin 1 (P8 – P2’), so why was it used?

The authors state in the abstract that RVC has a “relatively strict preference for Glycine at P3’ and P4’ “. This does not tally with their new ‘pseudo' cleavage of angiotensin 1, where the cleavage at Tyr4 has P3’,P4’ of Pro and Phe and the cleavage at Phe8 has a P3’,P4’ of Val and Asp.

There is no discussion of this, and these anomalies might suggest that the whole study is somewhat artifactual! 

In addition the fact that Ang l does not actually have a P3’ or P4’ when converted to Ang ll seems to be ignored.

As before, the relevance of the study to the angiotensin system falls short of what is required and should be addressed in a more comprehensive manner.

Note: there is quite a few grammatical errors, and some altered sentences (e.g. in the abstract) were more correct in the original manuscript.

Author Response

Dear Reviewer,

We have now rerun the entire section of the Ang I cleaving activity. Finally after trying to get  MS analysis with traditional MALDI-MS for more than one and a half year we finally convinced Alphalyse in Denmark to restart their service after having failed with several other MS providers. Most labs have closed down their MALDI- services and the new machines can apparently not offer this service, as several labs we contacted failed with their new machines.

Due to the very high price charged for this analysis we only could afford to run  four samples; Ang I uncleaved control, Ang I cleaved by the human chymase, Ang I cleaved by rMCP-1 and Ang I cleaved by RVC. The result shows that the human chymase only cleaves after Phe8 as expected, resulting in Ang II generation, whereas rMCP-1 cleaves at both Tyr4 and Phe8 as previously also shown leading to both generation and degradation of Ang II. Interestingly RVC was found to be a very poor angiotensin converter. Substantial cleavage of Ang I was first seen after adding more than 200 times more enzyme compared to cleavage of the most preferred site identified by phage display and then the Ang I was not primarily cleaved at Phe8 but at Arg2.

We did then confirm this by a new type of TRX substrates where we expose Ang I in the C-terminal end of the fusion protein preceded by a Ser-Gly kinker region and a His tag. This type of substrate was found to give identical results as the peptide-MS analysis with the three enzymes studied with both technologies. We therefore increased the analysis using this new type of substrate on a larger panel of mast cell chymases including hamster, opossum and platypus to study the more general aspects of chymase in blood pressure regulation.

                      The major finding of these two types of analysis was that they were remarkably consistent and showed that RVC is a very poor angiotensin converter. These results indicate that RVC has no role in angiotensin conversion and that rMCP-1 also is a poor converter similar to the hamster, opossum and platypus chymases. The previous results using the 2xTRX substrates showing the strong activity at Phe8 by rMCP-1 was most likely correctly as started by you (Reviewer) that the additional C-terminal sequences originating from the linker and the second TRX made the new surrounding of the Phe8 into a much better substrate compared to its normal position two amino acids from the C-terminal end. However, as we had not succeeded in getting any MS provider performing peptide analyses for more than a year we took this as the best alternative, which proved no to give correct results. However, after using both MS and the new single Trx substrates we now feel 100% confident that the results presented in the manuscript is fully correct.

                      The question is now if there are other enzymes having such a function in these species or if ACE is the prime enzyme in these species. However our results on RVC now very clearly show that RVC is not the enzyme responsible for the blood pressure regulating effect seen in these previous studies.

The language has been checked a second time by a native English speaking person to ensure correct language.

Round 3

Reviewer 2 Report

The authors have now done a substantial amount of additional experimentation to address earlier issues.
They have examined cleavage of the actual Ang1 peptide, both free peptide and as an exposed C-terminal of the Trx fusion. Although the results suggest RVC is not an angiotensin converting enzyme, contradicting previous reports based on indirect evidence, this fact makes a significant contribution to the field. It also demonstrates that phage display library specificities do not necessarily translate to physiological substrates.
Thus the manuscript should be accepted for publication, with minor revision as below, not subject to further review.

One anomaly which requires clarification: The authors suggest that cleavage of the Arg chromogenic substrate is likely due to contamination (carry forward) of enterokinase added for RVC activation. However they subsequently demonstrate cleavage after Arg2 in mass spec analysis of Ang1 digestion and assume this is RVC activity…could this also due to enterokinase? given that 200 fold higher enzyme was used? please comment

The source of the Ang1 peptide should be given in the Materials and Methods.

There are quite a few grammatical errors in the manuscript.

Author Response

We have now addressed all issues that were put forward by of the reviewer.

1. We have analysed the potential cleavage by Enterokinase on AngI with the same amount of enzyme that was present in the RVC analysis and seen a 50% cleavage at Arg2 by the enterokinase showing that at least a part of the Arg2 cleavage originate from enterokinase. This information has now been added to both the results and the discussion sections. A very good observation by the reviewer.

2. We have  added a section on describing the origin of the the Ang I peptide used in the MS analysis in the Materials and methods.

3. We have also once again gone through the text and removed a number of grammatical errors.

All the text changes made have been marked in red.

We have also made several esthetic changes due to the large gaps  on the pages in the proof we received. We have moved text to remove the large blank areas and make the manuscript more esthetic. Hope that you keep this new layout. We have carefully ensured that all the original text and figures are as in the type set manuscript you sent us.